# Chrysoprase color grading with machine learning: A systematic approach

Yuansheng Jiang[1,2], Ying Guo[1]*, Vien Cheung[2]*, Pohsun Wang[3], Stephen Westland[2]

**1** School of Gemmology, China University of Geosciences (Beijing), Beijing, China, **2** School of Design, University of Leeds, Leeds, West Yorkshire, United Kingdom, **3** Faculty of Innovation and Design, City University of Macau, Macau, China

* 2001011459@cugb.edu.cn (YG); T.L.V.Cheung@leeds.ac.uk (VC)

## Abstract

The color of gemstones plays a pivotal role in determining their quality and significantly impacts their market value. However, inconsistencies in gemstone color evaluation, stemming from the subjective nature of color perception, have hindered standardization in the market. Chrysoprase, celebrated for its distinctive apple-green color, is no exception. To address these challenges, this study employs a machine learning-based approach to automate chrysoprase color grading. CIE (1976) $L^*a^*b^*$ data were measured for 51 chrysoprase samples and 676 green reference points generated using the GemDialogue Color Reference using an X-Rite SP62 spectrophotometer. K-means was applied for color clustering, with Fisher discriminant analysis used to validate the clustering results. Various machine learning algorithms, including logistic regression, neural networks, k-nearest neighbors, support vector machines, and random forest, were trained on labeled data to assign chrysoprase colors to different groups. Logistic regression and neural network achieved comparably high macro F1-scores, and logistic regression was ultimately selected due to its simplicity, interpretability, and computational efficiency. In independent evaluation on 51 real chrysoprase samples, all samples were correctly classified within the present dataset. Additional mixed cross-validation incorporating both synthetic and real samples yielded consistent performance (macro F1-score = 99.59%), further supporting the robustness of the proposed approach. These results demonstrate the feasibility of applying machine learning techniques to structured gemstone color grading. The proposed framework provides a reproducible approach for objective chrysoprase color evaluation and may be adaptable to other gemstones with comparable colorimetric characteristics, subject to further validation. A publicly accessible chrysoprase color grading application is available at: https://github.com/harden2009190006/Chrysoprasecolorclassifier.

**Data availability statement:** All relevant data are within the manuscript and its Supporting Information files.

**Funding:** Y.J. received funding from the China Scholarship Council (CSC), Grant No. 202306400113. Funder: China Scholarship Council Funder website: https://www.csc.edu.cn The funder had no role in the study design, data collection and analysis, decision to publish, or preparation of the manuscript.

**Competing interests:** The authors have declared that no competing interests exist.

## Introduction

Chrysoprase is a cryptocrystalline variety of polycrystalline quartzite with a distinctive apple-green hue. Typically mined from the weathered nickel-bearing caps of serpentinites, chrysoprase deposits are predominantly associated with silicified zones of serpentinized ultramafic and mafic rocks [1–5]. In some instances, these deposits may also be linked to surrounding granitic formations [6]. This gemstone is found across the globe, in countries such as Australia [7], Tanzania [1,8], Poland [9], Kazakhstan, South Africa, Brazil, and the United States, among others [10]. Among these, Australia is renowned for producing the finest quality chrysoprase.

The apple-green color of chrysoprase is a hallmark characteristic, distinctly different from the green hues of chalcedonies colored by $Cr^{3+}$ or $Fe^{2+}/Fe^{3+}$ ions [3]. Although earlier hypotheses suggested that the coloration of chrysoprase was due to Fe, Co, or Cu, it is now well-established that the color is directly linked to its nickel (Ni) content, which ranges from a few tenths of a percent to several percent by weight [11,12]. Variations in Ni content also influence the chroma of chrysoprase [13], allowing it to form a series of gradually transitioning green colors.

The unique apple-green color of chrysoprase underscores the importance of color as a key factor in evaluating colored gemstone quality. While colored gemstone quality is typically assessed based on attributes such as color, clarity, and cut, color often takes precedence. As the most visually striking characteristic, it greatly enhances a gemstone's aesthetic appeal and serves as a major determinant of its overall value. However, the inherently subjective nature of color perception often leads to inconsistencies in gemstone color evaluation, resulting in a lack of standardization and regulatory frameworks in the market. Addressing this issue has been a long-standing priority for gemologists, who seek objective and reproducible methods for color assessment. In recent years, color science has played an increasingly pivotal role in gemmology, offering critical insights into the analysis and characterization of gemstones. Research in this field has extensively focused on a variety of gemstones, including color-change garnet [14–17], tourmaline [18,19], sapphire [20], alexandrite [21], peridot [22,23], cubic zirconia [24], blue amber [25], jadeite [26–30], citrine [31], amethyst [32], turquoise [33] and spinel [34]. For green gemstones, such as jadeite [30] and peridot [22,35], researchers have attempted to establish color grading systems to provide more structured evaluation criteria. Nonetheless, these grading approaches are primarily based on clustering the colors within a specific sample set, where categories are determined solely from the characteristics of that dataset. As a result, these methods are limited in their applicability and may not generalize well when grading new, untested samples. Therefore, using chrysoprase with its unique apple-green as a case study, developing a practical model to grade the color of new, untested samples holds great significance.

Machine learning is a transformative approach for developing self-learning systems that can efficiently analyze and interpret large datasets [36,37]. Machine learning can be categorized into unsupervised and supervised learning [38]. Unsupervised learning uses unlabeled datasets to identify patterns and clusters in data, while supervised learning utilizes labeled datasets to train algorithms for classification. In

recent years, machine learning has significantly impacted mineralogy and color science. For instance, in mineralogy, it has enabled the differentiation of quartz-forming environments [39], classification of detrital zircon source rocks [40], and identification of emerald origins [41,42]. In color science, machine learning has been used to characterize color cameras to enable them to make colorimetric measurements [43], categorize pepper seeds using color filter array images [44], extract colour palettes from fashion runway images [45] and predict visual similarity between color palettes [46]. These examples highlight the versatility of machine learning in addressing complex challenges across various disciplines.

This study addresses the challenge of grading chrysoprase colors using machine learning techniques. By integrating machine learning methods with colorimetric data, we propose a systematic approach for chrysoprase color grading based on CIE L*, a*, and b* values. K-means clustering is first applied for initial grouping, followed by validation using Fisher Discriminant Analysis. To ensure precision and reliability, multiple machine learning algorithms were tested, and the best-performing algorithm was selected to establish the final color grading model. The findings demonstrate the potential of machine learning-driven approaches in gemology, providing a robust and objective solution for gemstone color evaluation.

## Materials and methods

### Samples

A total of 51 natural chrysoprase samples from Australia were selected for this study, exhibiting a color range from pale to strong green. Of these, 41 samples were cut and polished into 7 mm diameter cabochons, while the remaining ten were shaped into 10 mm by 15 mm oval cabochons. The gemological properties of the chrysoprase samples are summarized in Table 1. Examples of these samples are shown in Fig 1.

**Table 1. The gemological characteristics of the chrysoprase samples.**

| Color | Magnified Observation | Refractive Index | Relative Density | UV Fluorescence | Chelsea Color Filter |
|---|---|---|---|---|---|
| yellow-green, green, blue-green | cryptocrystalline structure | 1.52-1.54 | 2.57-2.63 | Inert | Inert |

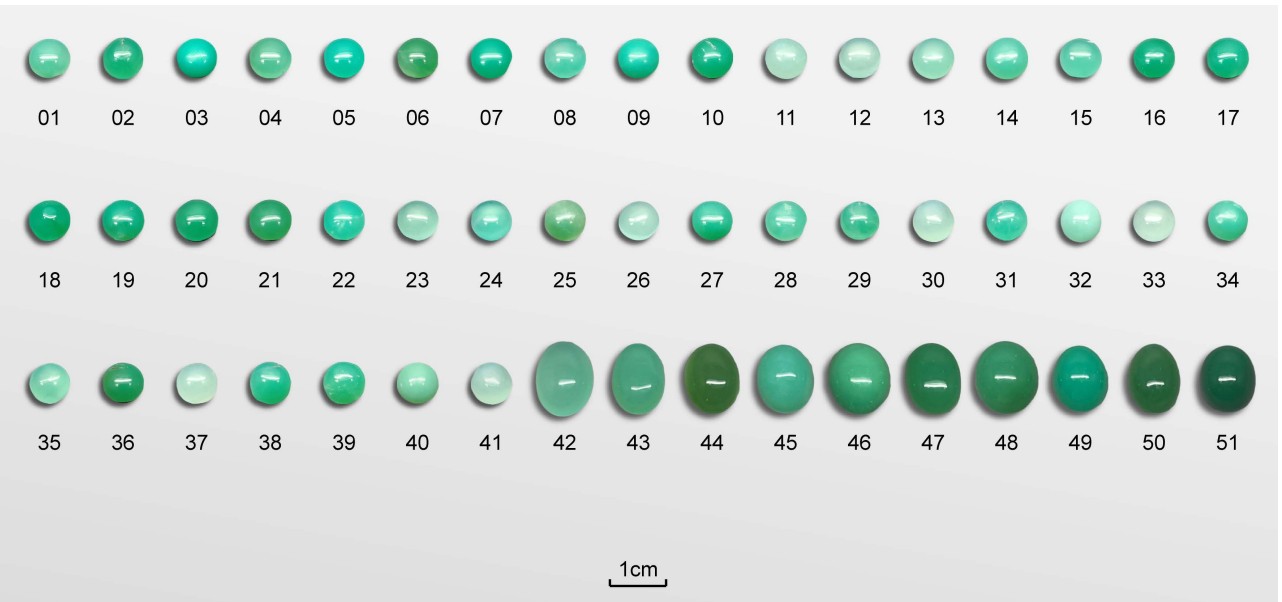

**Fig 1. Photograph showcasing the chrysoprase samples used in this study.**

## GemDialogue color chart

The GemDialogue color chart (https://www.gemdialogue.com/) is a commercially available color-matching tool specifically designed for gemology and the jewelry industry. It includes a series of transparent color cards that cover a range of hues, from red and blue to green and yellow. By layering different color masks, subtle color variations can be achieved (Fig 2).

To simulate the color of chrysoprase, this study selected six primarily green color standards from the GemDialogue chart: B3G (strong bluish green), B2G (moderate bluish green), B1G (slightly bluish green), G (green), Y1G (slightly yellowish green), and Y2G (moderate yellowish green). Additionally, three types of masks were used: transparent black/gray, transparent brown, and opaque black/white. Each GemDialogue color chart and mask consists of ten uniform color strips, which vary progressively in both lightness and chroma, allowing for systematic overlay testing. Among the selected six green-dominant color standards, B2G, G, and Y2G are single-layered base standards, while B3G, B1G, and Y1G are composite standards created by overlaying two base standards. For the three single-layered color standards—B2G, G, and Y2G—five overlay methods were applied with the three masks (Fig 3a): (1) placing the black/gray mask above the color standard, (2) placing the black/gray mask below the color standard, (3) placing the brown mask above the color standard, (4) placing the brown mask below the color standard, and (5) placing the black/white mask below the color standard. Each single-layered color standard thus produces 510 color data points, calculated as follows: 10 original

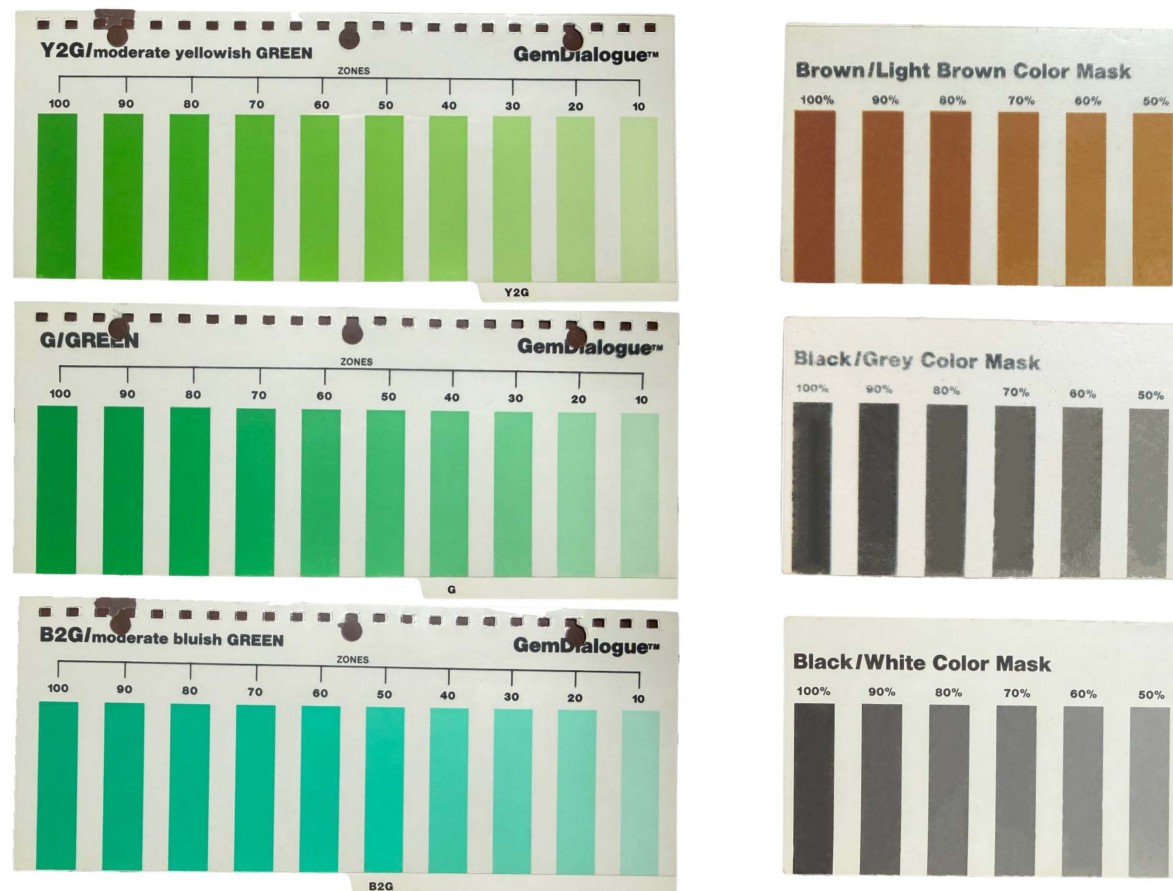

**Fig 2. GemDialogue color charts used in the study.** The left side displays GemDialogue color standards for B2G (moderate bluish green), G (green), and Y2G (moderate yellowish green); the right side shows color masks for Black/Grey, Brown, and Black/White.

                                                                  

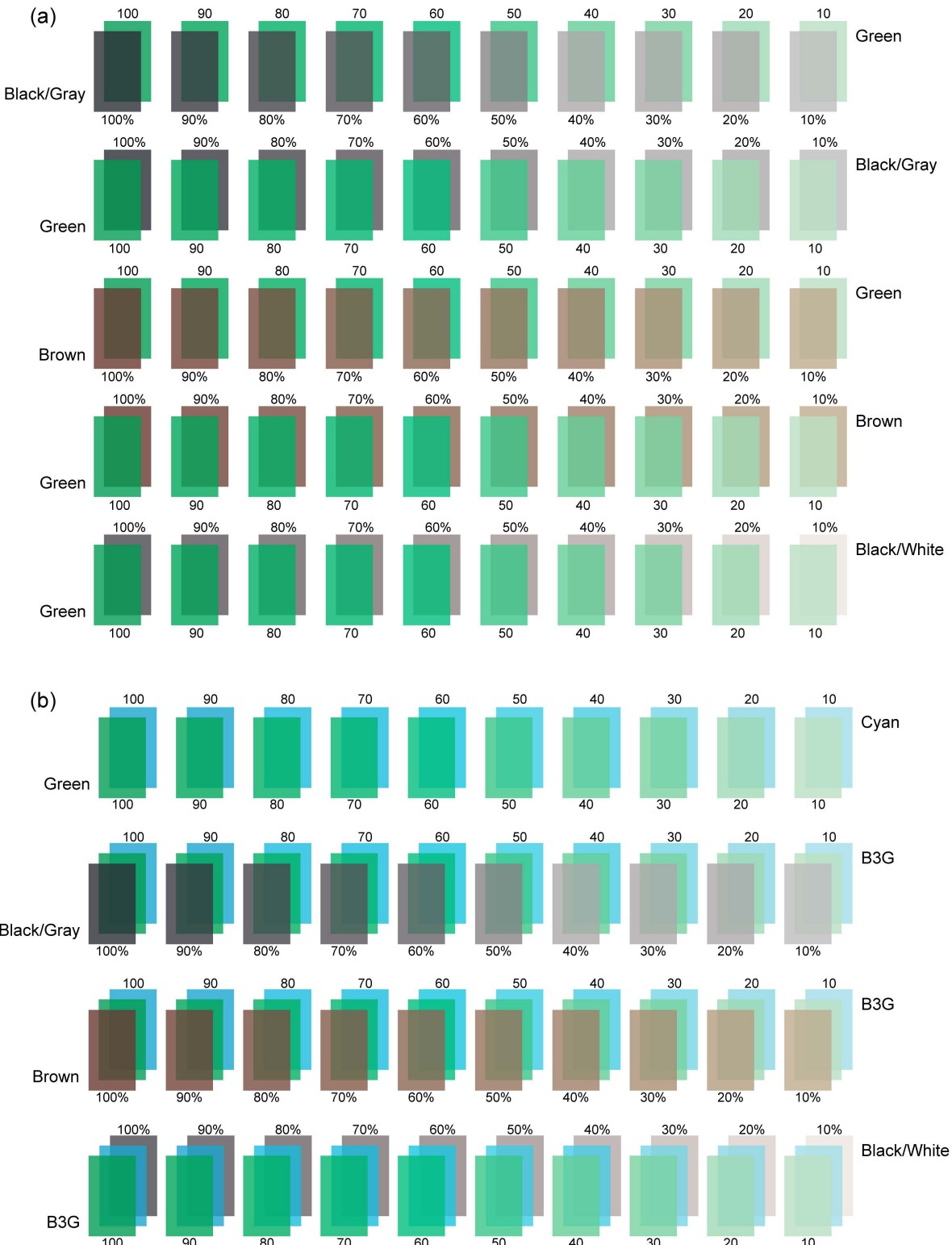

**Fig 3. Overlaying Methods of GemDialogue Color Chips. (a)** Single color standard overlays using G as an example: (1) black/gray mask above the color standard, (2) black/gray mask below the color standard, (3) brown mask above the color standard, (4) brown mask below the color standard, and

(5) black/white mask below the color standard. **(b)** Composite color standard overlays using B3G as an example: The first row shows composite color standards created by overlaying single-layered base standards with no mask applied. In subsequent rows, composite standards were generated by aligning and overlapping corresponding color strips with matching lightness and chroma. Masks were then applied using three methods: (1) black/gray mask above, (2) brown mask above, and (3) black/white mask below the composite color standard.

color strips (without any mask overlay) + 5 overlay methods × 10 color strips × 10 mask strips. With three single-layered color standards (B2G, G, and Y2G), a total of 1,530 color data points were obtained.

Composite color standards were created by overlaying single-layered base standards. Specifically, the B3G color standard was formed by overlaying G onto Cyan, the B1G color standard by overlaying B2G onto G, and the Y1G color standard by overlaying Y onto Cyan (Fig 3b).

Without the use of masks, composite color were generated by overlaying any color strip from one base standard with any color strip from the other base standard. This approach produced 10 × 10 = 100 unique color data points for each composite standard. As the GemDialogue color standards are partially transparent but not fully transparent, the resulting composite colors had reduced brightness compared to individual single-layered standards due to the overlapping layers.

When masks were applied, the composite color standards (B3G, B1G, and Y1G) were first created by fully aligning and overlapping corresponding color strips with the same degree of lightness and chroma. As illustrated in Fig 3b, three mask overlay methods were applied: (1) placing the black/gray mask above the composite color standard, (2) placing the brown mask above the composite color standard, and (3) placing the black/white mask below the composite color standard. For each composite color standard, this resulted in a total of 400 color data points, calculated as 100 (without masks) + 3 overlay methods × 10 color strips × 10 mask strips. With three composite color standards (B3G, B1G, and Y1G), a total of 1,200 color data points were obtained. Combined with the 1,530 data points derived from the three single-layered base standards, the study produced 1,530 + 1,200 = 2,730 color data points from six standards.

### CIE (1976) L*a*b*

The CIE (1976) $L^*a^*b^*$ uniform color space aligns with human visual perception by offering (1) high color uniformity, where the perceived color differences are proportional to the Euclidean distance between color coordinates, and (2) adherence to the subjective principle that visual sensitivity to color differences is lower in the red–green direction than in the yellow–blue direction [47]. CIELAB is structured as a three-dimensional spherical color space, characterized by colorimetric coordinates ($a^*$ and $b^*$) in the horizontal plane and lightness $L^*$ along the vertical axis [48]. Lightness $L^*$ typically ranges from black ($L^* = 0$) to white, with higher values indicating increased lightness [49]. The coordinate $a^*$ represents variations from red ($+a^*$) to green ($-a^*$), while $b^*$ represents transitions from yellow ($+b^*$) to blue ($-b^*$) [50]. Chroma $C^*$ ranges from the weakest ($C^* = 0$) to the strongest. A higher chroma value corresponds to greater brilliance and intensity [49]. The hue angle h°, ranging from 0 to $2\pi$, signifies a continuous progression of hues, including red, yellow, green, blue, and purple [50]. All color parameters are psychophysical and have no units. Chroma $C^*$ and hue angle h° can be calculated from $a^*$ and $b^*$ coordinates as follows:

$$C^* = \sqrt{a^{*2} + b^{*2}} \tag{1}$$

$$h° = arctan\frac{b^*}{a^*} \tag{2}$$

### Color measurement

Reflectance spectra were measured using a portable X-Rite SP62 spectrophotometer (X-Rite, Grand Rapids, MI, USA) with d/8° optical geometry in SCI mode (specular component included). The spectral range was 400–700 nm with a 10 nm interval, and the measurement aperture was 5 mm.

All specimens were polished cabochons. Measurements were performed at a single standardized location, the apex (top center) of the domed surface. The instrument was positioned perpendicular to the surface in contact mode, with samples placed on a Munsell Neutral Value N9.5 background card [51].

Prior to measurement, the instrument was calibrated using the manufacturer's white calibration standard (ceramic tile) and black trap. For each specimen, three consecutive measurements were taken at the same location and averaged. According to the manufacturer's specification, the short-term repeatability of the instrument is typically within $\Delta E^* \leq 0.04$.

CIELAB values were calculated under illuminant D65 with the 2° standard observer. Surface curvature and polishing of cabochon gemstones may influence reflectance measurements due to geometric effects. To reduce such variability, measurements were consistently taken at the apex of the cabochon surface and SCI mode (specular component included) was used so that specular reflection from the polished surface was included in the measurement. Although minor geometric influences cannot be completely eliminated for curved gemstone surfaces, the measurement conditions were standardized across all samples to ensure comparability of the colorimetric data. The reported color values therefore represent the optical appearance of polished cabochon specimens measured under these standardized conditions.

## Unsupervised learning

K-means clustering is an unsupervised learning algorithm designed to partition data into clusters, with each data point assigned to the cluster whose mean (centroid) is nearest to it. The algorithm iteratively refines cluster centroids by minimizing the within-cluster variance, following four main steps: (1) initializing cluster centers. (2) assigning samples to the nearest cluster. (3) recalculating cluster centroids. (4) repeating the process until convergence is achieved [52–54].

In this study, k-means clustering was implemented using MATLAB R2024a to group and label the color data into distinct clusters, enabling an objective grouping based on inherent color characteristics. The Euclidean distance metric was used to calculate the distances between data points. To avoid inconsistencies from the random initialization of centroids, the 'Replicates' parameter was set to 1,000, running k-means 1,000 times to obtain the optimal clustering result. In the k-means algorithm, the number of clusters must be predefined. To evaluate the clustering quality under different numbers of clusters, silhouette values were calculated. These values, ranging from −1 to +1, indicate how well an observation aligns with its assigned cluster compared to other clusters. The overall cluster quality was assessed using the average silhouette value, which represents the mean silhouette value of all data points. Higher average silhouette values suggest better-defined clusters, making clustering schemes where most observations have high silhouette values more desirable.

## Fisher discriminant analysis

Fisher Discriminant Analysis (FDA) is a method that utilizes variance analysis to establish a discriminant function for rapid classification of datasets. The basic process involves constructing the discriminant function based on known class data. FDA identifies one or more directions in the original sample space to project the samples. It determines the coefficients of the discriminant function by maximizing the between-class covariance and minimizing the within-class covariance, thereby identifying the optimal projection direction. Next, a distance-based criterion is used to distinguish samples from different classes. Finally, the discriminant scores are calculated and compared with a threshold to assign the samples to their respective classes. The advantage of FDA lies in its broad applicability, as it does not impose strict requirements on the data distribution and variance, making it highly effective for classifying complex objects [55–57]. FDA was performed in SPSS 25 software to assess the quality of the clustering results.

## Supervised learning

**Model construction.**  The labeled data were divided into two subsets: one comprising color data generated by overlaying GemDialogue color chips, designated as the training set, and the other containing color data from 51 actual chrysoprase samples, designated as the test set. The training set was utilized for model development and parameter tuning, while the test set was reserved for evaluating the model's performance [58]. Five well-established supervised

learning algorithms were applied to build chrysoprase color grading models using MATLAB R2024a with labeled color data. The algorithms include logistic regression (LR), which predicts class probabilities through a sigmoid function and performs effectively with data that are linearly separable [59]. Neural networks (NN) mimic human neural processing, leveraging layers of interconnected nodes to model intricate, non-linear patterns in data [60]. Support vector machines (SVM) excel in classification tasks by identifying an optimal hyperplane to distinguish classes, utilizing kernel functions to address non-linear relationships [61]. K-nearest neighbors (KNN) assigns an outcome based on the dominant class among a data point's nearest neighbors, offering an intuitive but computationally intensive method [62]. Lastly, Random Forest (RF) constructs an ensemble of decision trees through bootstrap aggregation and random feature selection. By averaging predictions across multiple decorrelated trees, RF effectively reduces variance and overfitting, resulting in strong generalization performance on complex datasets [63,64]. These methods were selected to provide a well-rounded assessment of chrysoprase color grading capabilities. More importantly, these algorithms are widely used in classification problems within geoscience, gemology, and colorimetry [39,40,42–46,65].

**Model selection.** Testing models on unseen data is crucial for evaluating the generalization ability of supervised machine learning models. Typically, this is achieved by withholding a portion of the training set and using the withheld data to assess model performance [66]. In this study, a five-fold cross-validation technique was employed for model selection and hyperparameter optimization. The dataset from the training set (color data generated by overlaying GemDialogue color chips) was randomly divided into five folds. In each iteration, four folds were used for training, while the remaining fold served as the validation set, which was used to assess model performance and guide hyperparameter tuning. This process was repeated five times, ensuring that each fold was used once as a validation set. The final performance result was obtained by aggregating the total number of correct and incorrect classifications across all folds, providing a comprehensive evaluation metric. Compared to a simple train/test split, this approach reduces the risk of high bias and improves the model's generalization capability on unseen data. After comparing the performance of five algorithms, the best-performing algorithm was selected, and the model was retrained using the entire training set. The testing set (color data from 51 actual chrysoprase samples), which was completely excluded from the training process, was then used to further evaluate the model's performance.

Several metrics can be used to select the optimal classifier. The most commonly used metric is accuracy, which simply measures the proportion of correct predictions made by the classifier. However, accuracy becomes less informative when dealing with class-imbalanced datasets or multi-class classification problems. In such cases, additional metrics provide a more comprehensive assessment of model performance. The true positive rate (TPR), also known as sensitivity or recall, quantifies the proportion of actual positive observations that were correctly classified:

$$TPR = \frac{TP}{TP + FN}$$

$$(3)$$

where TP represents true positives, and FN represents false negatives.

The precision (PPV) measures the proportion of correctly classified positive observations among all instances predicted as positive:

$$PPV = \frac{TP}{TP + FP}$$

$$(4)$$

where FP represents false positives.

The F1 Score is the harmonic mean of TPR and PPV, offering a balanced measure that accounts for both false positives and false negatives:

$$F1\ score = 2 \times \frac{TPR \times PPV}{TPR + PPV}$$

$$(5)$$

 

In this study, the macro F1-score of the validation set, computed using five-fold cross-validation, was employed to select the optimal algorithm. The macro F1-score calculates the F1-score for each class independently and then averages the results, ensuring that all classes are given equal importance, regardless of their sizes.

**Hyperparameter optimization.** Hyperparameters are parameters that are not directly learned during machine learning but instead control how the algorithm constructs the model and learns from the data. These parameters are specified prior to training during model initialization. Examples of hyperparameters include the number of neighbors and distance metric in k-nearest neighbors (KNN), the number and size of layers in neural network (NN), the regularization strength in logistic regression (LR), the number of trees and maximum tree depth in a random forest (RF), and the regularization factor and kernel function in a support vector machine (SVM).

To determine the optimal hyperparameters for each algorithm, Bayesian optimization [67] was performed within MATLAB R2024a Classification Learner using the default configuration (acquisition function: Expected Improvement per Second (plus), maximum objective evaluations: 30). Five-fold cross-validation was employed during the optimization process, and the macro F1-score was used as the objective function for model selection. The hyperparameter optimization ranges for the five algorithms are summarized in S2 Table in the Supplementary Material. To ensure reproducibility, a fixed random seed (rng(123,'twister')) was used in all experiments, making the training and evaluation procedures deterministic. Fig 4 illustrates the detailed workflow of our approach.

**Fig 4. Workflow for model development.** Five classical machine learning algorithms were first constructed using the training dataset. Hyperparameter tuning was performed via Bayesian optimization with five-fold cross-validation, using the macro F1-score as the optimization objective. The model with the highest cross-validated macro F1-score was selected as the optimal model. Finally, an independent testing set consisting of 51 chrysoprase samples was used to evaluate the model's performance.

### CIEDE2000 color difference

To evaluate the perceptual similarity between measured chrysoprase samples and the synthetic color dataset, the CIEDE2000 color difference ($\Delta E_{00}$) was employed. Compared with earlier color-difference formulas, the CIEDE2000 metric improves perceptual uniformity by introducing correction terms for lightness, chroma, and hue interactions.

In this study, $\Delta E_{00}$ values were calculated between each measured chrysoprase sample and all synthetic color points. For each sample, the minimum $\Delta E_{00}$ was determined using a nearest-neighbour approach to assess how closely the synthetic dataset approximates the empirical chrysoprase color distribution.

The $\Delta E_{00}$ values were computed according to the formulation proposed by Sharma et al. [68].

### Inclusivity in global research

This study did not involve human participants, animals, Indigenous communities, or field sampling. All chrysoprase samples were analyzed using laboratory-based colorimetric measurements under controlled laboratory conditions. Therefore, no ethical approval or governmental permits were required. Additional information regarding ethical, cultural, and scientific considerations related to inclusivity in global research is provided in the Supporting Information (S1 Checklist).

## Result and discussion

### Acquisition of color data

**Color data from chrysoprase samples.** The range of CIELAB values for the 51 chrysoprase samples were lightness $L^*$ (38.5 to 76.14), $a^*$ (−44.67 to −14.7), $b^*$ (0.68 to 21.79), chroma $C^*$ (18.20 to 46.96) and hue angle $h°$ (143.22 to 178.40). These values align well with the color appearance of chrysoprase. The colors of the 51 samples are displayed in Fig 5.

**Color data preprocessing.** A total of 2730 green-dominant color data sets were obtained by overlaying the GemDialogue color chips. However, some colors produced were too dark or too light, with certain samples exhibiting excessively high or low chroma, while others appeared too yellowish or bluish.

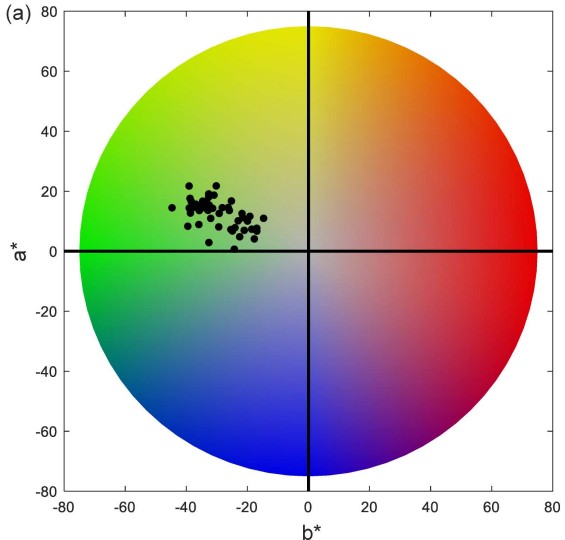 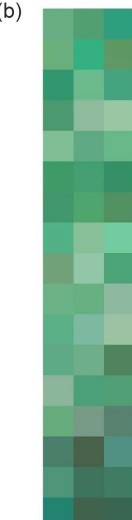

**Fig 5. Color characteristics of the 51 chrysoprase samples. (a)** The color distribution of 51 chrysoprase samples plotted in the CIELAB color space. **(b)** The corresponding visual representation of the samples' colors.

To ensure that the synthetic dataset represents the empirically observed chrysoprase color domain, a filtering procedure was applied based on the colorimetric bounds derived from the 51 measured chrysoprase samples. Chroma C* and hue angle h° were calculated from the CIELAB coordinates according to Eqs. (1) and (2).

The measured chrysoprase samples exhibited the following ranges: lightness L* from 38.50 to 76.14, chroma C* from 18.20 to 46.96, and hue angle h° from 143.22° to 178.40°. A synthetic color point was retained only if it simultaneously satisfied all three constraints:

$$L^* \in \left[L^*_{min}, L^*_{max}\right] \;\wedge\; C^* \in \left[C^*_{min}, C^*_{max}\right] \;\wedge\; h^{\circ} \in \left[h^{\circ}_{min}, h^{\circ}_{max}\right] \tag{6}$$

This strict filtering procedure reduced the dataset from 2,730 overlay-generated color points to 676 representative synthetic color points. The filtering ensures that the synthetic dataset remains within the empirical color domain defined by the measured chrysoprase samples, thereby preventing extrapolation beyond the observed chrysoprase color range.

The resulting synthetic dataset therefore acts as a dense sampling grid within this empirical color domain, improving the continuity of the color-space representation without introducing additional class structures.

It should be acknowledged that GemDialogue overlays simulate color through planar transmissive stacking and therefore cannot fully reproduce the optical behaviour of natural chrysoprase cabochons, such as subsurface scattering in microcrystalline quartz, specular reflection from curved surfaces, and inclusion-related heterogeneity. In the present study, the overlays are used as a controlled method to densely sample the perceptually relevant region of CIELAB color space rather than to physically model gemstone optics.

**Validation of synthetic color coverage.** To assess whether the filtered synthetic dataset adequately covers the empirical chrysoprase color domain, a nearest-neighbour analysis was performed using the CIEDE2000 color difference formula ($\Delta E_{00}$). For each measured chrysoprase sample ($n = 51$), the minimum $\Delta E_{00}$ relative to the synthetic dataset ($n = 676$) was calculated.

The results show that 47.06% of measured samples have a synthetic counterpart within $\Delta E_{00} < 1$, which is commonly regarded as approximately the perceptibility threshold for human observers [68,69]. Furthermore, 88.24% of samples fall within $\Delta E_{00} < 2$, a tolerance range commonly adopted in industrial color evaluation [70]. The median minimum $\Delta E_{00}$ was 1.06, with the 95th percentile at 2.26.

These results indicate that the synthetic sampling grid provides substantial perceptual coverage of the empirical chrysoprase color domain and closely approximates the observed distribution of chrysoprase colors in CIELAB space.

## Labeling of color data

The k-means algorithm was employed to cluster a total of 727 sets of color data, each represented by L*, a*, and b* values. Among them, 676 sets were generated from a systematic combination of GemDialogue color chips, intended to comprehensively simulate the perceptual color space of chrysoprase. The remaining 51 sets were obtained from real chrysoprase samples. By combining both datasets, we aimed to create a more complete and representative color distribution for clustering.

The clustering served two main purposes: (1) to establish perceptually meaningful color categories that comprehensively cover the green color space associated with chrysoprase, and (2) to assign each data point—whether derived from color chips or real samples—to one of these categories. The resulting cluster labels were then used as reference classes for subsequent supervised learning.

The next step was to determine the appropriate number of clusters. Since the 676 sets of color data generated from GemDialogue color chips were systematically constructed and evenly distributed in the color space, they did not exhibit natural clustering patterns. Consequently, the silhouette score—used to assess clustering quality by comparing intra-cluster cohesion and inter-cluster separation—may produce artificially high values at lower cluster numbers when applied

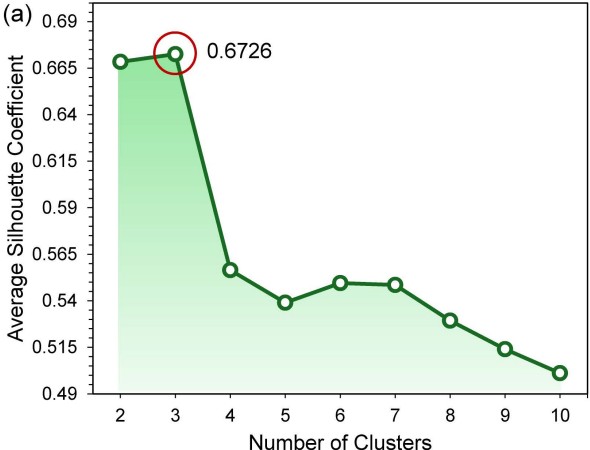 PLOS One

to uniformly distributed data. Indeed, when clustering the chip-based data alone, the silhouette analysis suggested that two clusters yielded the highest score. However, this result is mathematically driven and lacks perceptual or practical significance for chrysoprase color grading. Therefore, the number of clusters was determined using only the 51 real chrysoprase samples to anchor the clustering structure to the empirical color distribution of measured specimens. This approach implicitly assumes that the available real samples adequately represent the perceptual diversity of chrysoprase color. If certain color variations are under-represented in this dataset, the resulting cluster structure may not fully capture the broader population distribution. This limitation should therefore be considered when interpreting the clustering results.

To determine the optimal number of clusters in a perceptually meaningful way, silhouette values were calculated for cluster counts ranging from 2 to 10 based on their $L^*$, $a^*$, and $b^*$ values. As shown in Fig 6a, clustering the samples into three groups produced the highest average silhouette value (0.6726), indicating that three clusters best represent the natural distribution of chrysoprase colors.

Based on this result, we then applied k-means clustering with three clusters to the full dataset—including the 676 GemDialogue color chip data points and the 51 real chrysoprase samples (727 in total)—to assign each sample to a perceptually meaningful color group within the chrysoprase color space, thereby labeling the entire dataset for subsequent analysis. The centroids of the three clusters are listed in Table 2. To visualize the clustering results in two dimensions, principal component analysis (PCA) was applied to reduce the three-dimensional color data ($L^*$, $a^*$, $b^*$) to two principal components. The data points were then plotted according to their assigned cluster labels, as shown in Fig 6b.

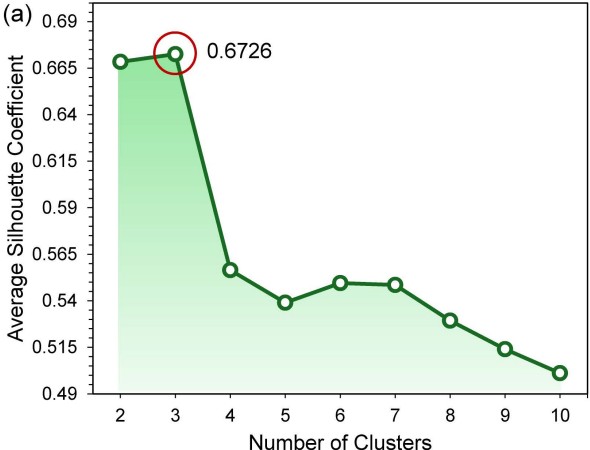
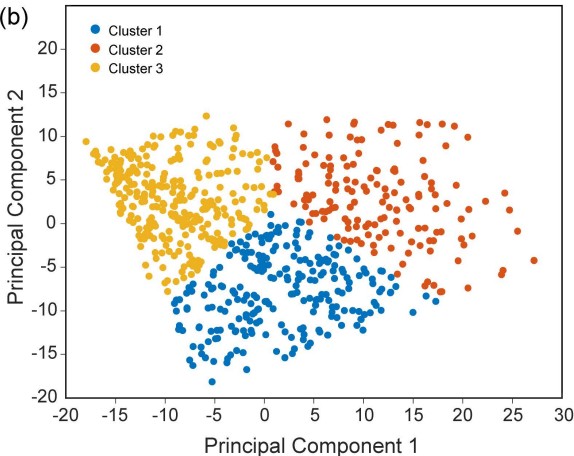

**Fig 6. Cluster determination and visualization in chrysoprase color data. (a)** Average Silhouette Coefficient for different numbers of clusters in k-means clustering. **(b)** 2D scatter plot of k-means clustering results, where PCA was performed on $L^*$, $a^*$, and $b^*$ color properties.

**Table 2. The centroids of the three clusters.**

| cluster center | $L^*$ | $a^*$ | $b^*$ | Simulated Color |
|---|---|---|---|---|
| 1 | 50.30 | −38.24 | 11.34 | |
| 2 | 63.75 | −29.60 | 9.30 | |
| 3 | 44.22 | −24.64 | 9.39 | |

## Validation of clustering results

To assess the k-means clustering results of the chrysoprase color data, Fisher Discriminant Analysis was employed, yielding three discriminant functions (D1 through D3) for clustering verification. These functions were constructed based on the color parameters $L^*$, $a^*$ and $b^*$, capturing the optimal projection direction for maximizing the distinction between color clusters. The discriminant formulae are as follows:

$$D1 = 1.808L^* - 1.297a^* + 0.249b^* - 72.768 \tag{7}$$

$$D2 = 2.382L^* - 0.903a^* + 0.141b^* - 91.045 \tag{8}$$

$$D3 = 1.627L^* - 0.790a^* + 0.192b^* - 47.712 \tag{9}$$

Each discriminant function is used to calculate a specific score, D1 through D3, which reflects the degree of association between a color data point and each color cluster. By substituting the values of $L^*$, $a^*$ and $b^*$ into these functions, a set of scores is generated for each data point. The data point is then assigned to the cluster corresponding to the highest score among D1 through D3, which indicates the cluster with the closest matching color characteristics.

The result of Fisher Discriminant Analysis are shown in Fig 7a. Among the 727 data points, five were misclassified. The accuracy is 99.31%, demonstrating the reliability of the clustering result. Compared to the previous chrysoprase study by Jiang and Guo (13), which applied k-means clustering to the color data of 41 chrysoprase samples, this study incorporates additional chrysoprase samples, thereby expanding the color range covered by the clustering analysis. Furthermore, to simulate other possible chrysoprase colors within this range, 676 representative chrysoprase color data points were

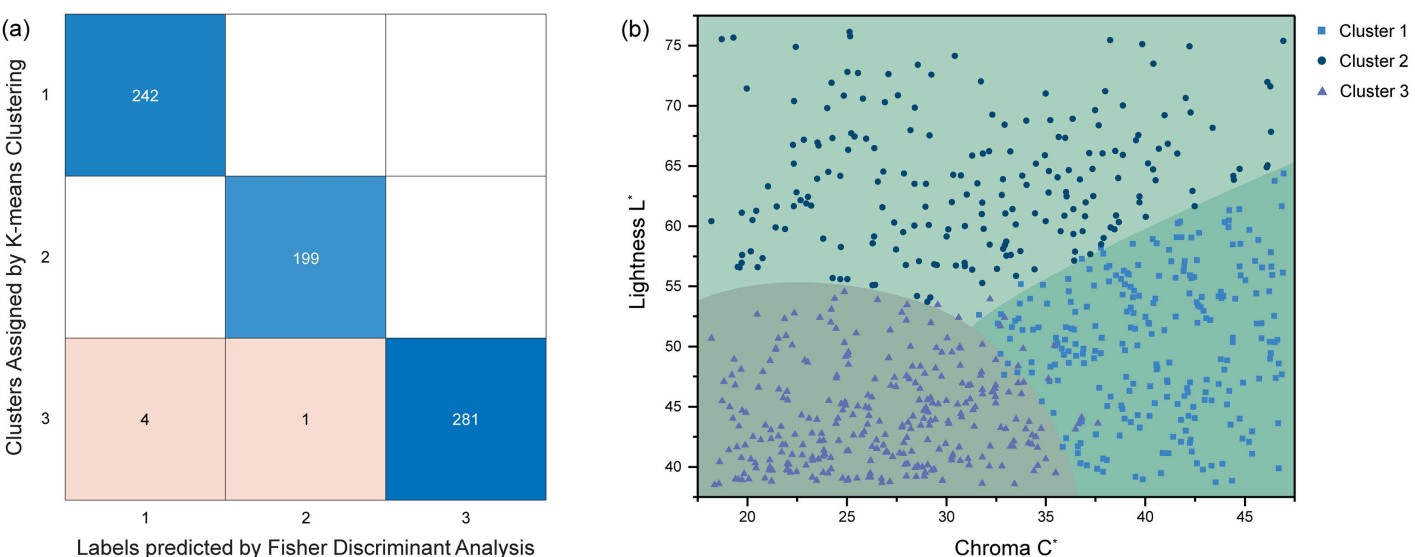

**Fig 7. Clustering validation and $L^*/C^*$ distribution of chrysoprase color data. (a)** Confusion matrix comparing k-means clustering assignments and Fisher Discriminant Analysis (FDA) predictions. The rows represent clusters assigned by k-means, while the columns indicate labels predicted by FDA. The numbers within the squares denote the sample counts in each category. **(b)** Distribution of Lightness $L^*$ and Chroma $C^*$ for chrysoprase clusters. The scatter plot represents 727 data points, grouped into three clusters with characteristic $L^*$ and $C^*$ values. The background shading approximates their boundaries.

generated using GemDialogue color chip overlays and included in the k-means clustering process. This enhancement further improves the generalizability and reliability of the clustering results compared to previous studies.

After identifying the three clusters, assigning meaningful names to each color category became essential for effective communication in the trade and identification of chrysoprase. Within the hue angle range of 143.22° to 178.40°, chrysoprase colors exhibit distinct $L^*$ and $C^*$ characteristics across the three clusters, as illustrated in Fig 7b.

To establish a systematic and widely recognized nomenclature, this study references the "Fancy grade" nomenclature of the Gemological Institute of America (GIA) [71], which is widely used in colored diamond grading. In this system, ranges of lightness and chroma values are grouped and described using a unified terminology.

Accordingly, the three clusters were named as follows: Cluster 1, characterized by high chroma and relatively low lightness, was designated as "Fancy Intense"; Cluster 2, characterized by relatively high lightness, was designated as "Fancy"; and Cluster 3, characterized by lower chroma and lower lightness, was designated as "Fancy Deep".

This nomenclature provides a structured and standardized approach for describing chrysoprase color variations, facilitating clearer communication in gemological research and commercial trade. It should be noted that this terminology is adopted here only as a descriptive reference for relative lightness–chroma relationships rather than as a direct application of the GIA diamond grading system to chrysoprase. Similar lightness–chroma descriptors have been used in gemstone color studies to facilitate systematic communication of perceptual color variation [22,32,72].

## Selection and construction of machine learning models

The labeled color data were divided into a training set and a testing set. The training set consists of 676 color data points, represented by color parameters $L^*$, $a^*$, $b^*$, generated using GemDialogue color chips. The testing set consists of color data, also represented by color parameters $L^*$, $a^*$, $b^*$, from 51 actual chrysoprase samples. In this study, the macro F1-score of the validation set, computed through five-fold cross-validation, was used for model selection. Within each fold, the training set was further split into a training subset and a validation set, with the latter used to assess model performance and guide hyperparameter tuning. This process was repeated across all five folds, ensuring that each subset served as a validation set once. A Bayesian optimization algorithm was employed to automatically tune the model's hyperparameters.

The macro F1-scores of the five algorithms, ranked in descending order, are neural network (99.70%), logistic regression (99.56%), support vector machine (98.82%), k-nearest neighbors (97.49%), and random forest (97.04%). Table 3 summarizes the results and lists the hyperparameters used for each algorithm. Confusion matrices of the five algorithms, derived from the validation sets in five-fold cross-validation, were used to illustrate the detailed classification results for each class (Fig 8). Among these, both logistic regression and neural network achieved comparably high macro F1-scores (99.56% and 99.70%, respectively). Although the neural network showed a marginally higher score, the performance difference (0.14%) is negligible. Considering the simplicity, interpretability, and lower computational complexity of logistic regression, we selected logistic regression as the final classification model.

The classification performance of the logistic regression model on the independent test set is illustrated in Fig 9. Fig 9a shows the decision regions of the classifier in PCA-reduced color space derived from the $L^*a^*b^*$ features. The independent chrysoprase samples are located well within their corresponding decision regions, indicating clear separation among the three color categories. Fig 9b presents the confusion matrix for the 51 independent chrysoprase samples.

The testing set, consisting of the color data ($L^*$, $a^*$, $b^*$) from 51 chrysoprase samples, was used to evaluate the logistic regression model, and all samples were correctly classified, yielding an overall accuracy of 100% (51/51). To quantify the statistical uncertainty associated with the limited number of real samples, a 95% confidence interval for the classification accuracy was computed using the Wilson score interval for binomial proportions [73]. The resulting interval is 93.0%–100%, indicating that although perfect classification was observed within the present dataset, the true underlying accuracy consistent with this sample size may be lower than 100%.

**Table 3. Optimal hyperparameters and macro F1 scores on the validation set for the five applied algorithms.**

| Algorithm | Hyperparameters | Value | Macro F1-score |
|---|---|---|---|
| **neural network** | number of hidden layers | 1 | 99.70% |
| | activation function | none | |
| | regularization strength ($\lambda$) | $5.123 \times 10^{-8}$ | |
| | standardization | yes | |
| | maximum iterations | 1000 | |
| | hidden layer size | 5 | |
| **logistic regression** | solver | lbfgs | 99.56% |
| | regularization | $L^2$ (ridge) | |
| | lambda | 0.0022 | |
| | Beta tolerance | $1 \times 10^{-4}$ | |
| | multiclass strategy | one-vs-one | |
| **support vector machine** | kernel function | quadratic | 98.82% |
| | kernel scale | 1 | |
| | box constraint level | 449.36 | |
| | multiclass coding | one-vs-one | |
| | standardize data | no | |
| **k-nearest neighbors** | number of neighbors (k) | 28 | 97.49% |
| | distance metric | minkowski (p=3) | |
| | distance weighting | inverse squared | |
| | standardization | no | |
| **random forest (bagged trees in MATLAB)** | ensemble method | bagging | 97.04% |
| | maximum splits | 601 | |
| | number of trees | 203 | |
| | number of variables sampled per split (mtry) | 1 | |
| | split criterion | gini (default) | |

Among the 51 real chrysoprase samples, the distribution across the three color categories was: Fancy Intense (n = 11), Fancy (n = 32), and Fancy Deep (n = 8). Because no misclassification occurred, all classes achieved precision, recall, and F1-score values of 1.00. The per-class metrics are summarized in Table 4. However, due to the limited number of samples in some categories (particularly Fancy Deep), the statistical uncertainty of these estimates remains relatively large, as reflected by the reported confidence intervals. Using the Wilson interval, the 95% confidence intervals for recall were 0.74–1.00 for Fancy Intense, 0.89–1.00 for Fancy, and 0.68–1.00 for Fancy Deep.

To further assess classifier confidence, posterior class probabilities were extracted from the logistic regression–based ECOC classifier for the independent test set. The predicted probabilities for the assigned classes ranged from 0.6626 to 1.0000 (mean = 0.9752, median = 0.9988), indicating generally high confidence in the model predictions.

To evaluate the potential domain mismatch between synthetic GemDialogue-derived color data and real chrysoprase samples, an additional stratified five-fold cross-validation was conducted on the combined dataset (n = 727), in which synthetic and real samples were randomly mixed during fold assignment. The logistic regression model achieved a macro F1-score of 99.59%, with three misclassifications across all folds. The performance obtained under the mixed cross-validation scheme remained highly consistent with that of the original external synthetic-to-real evaluation.

These findings indicate that the synthetic color space constructed in this study adequately represents the perceptual distribution of real chrysoprase colors. The consistently high performance across evaluation schemes suggests that the

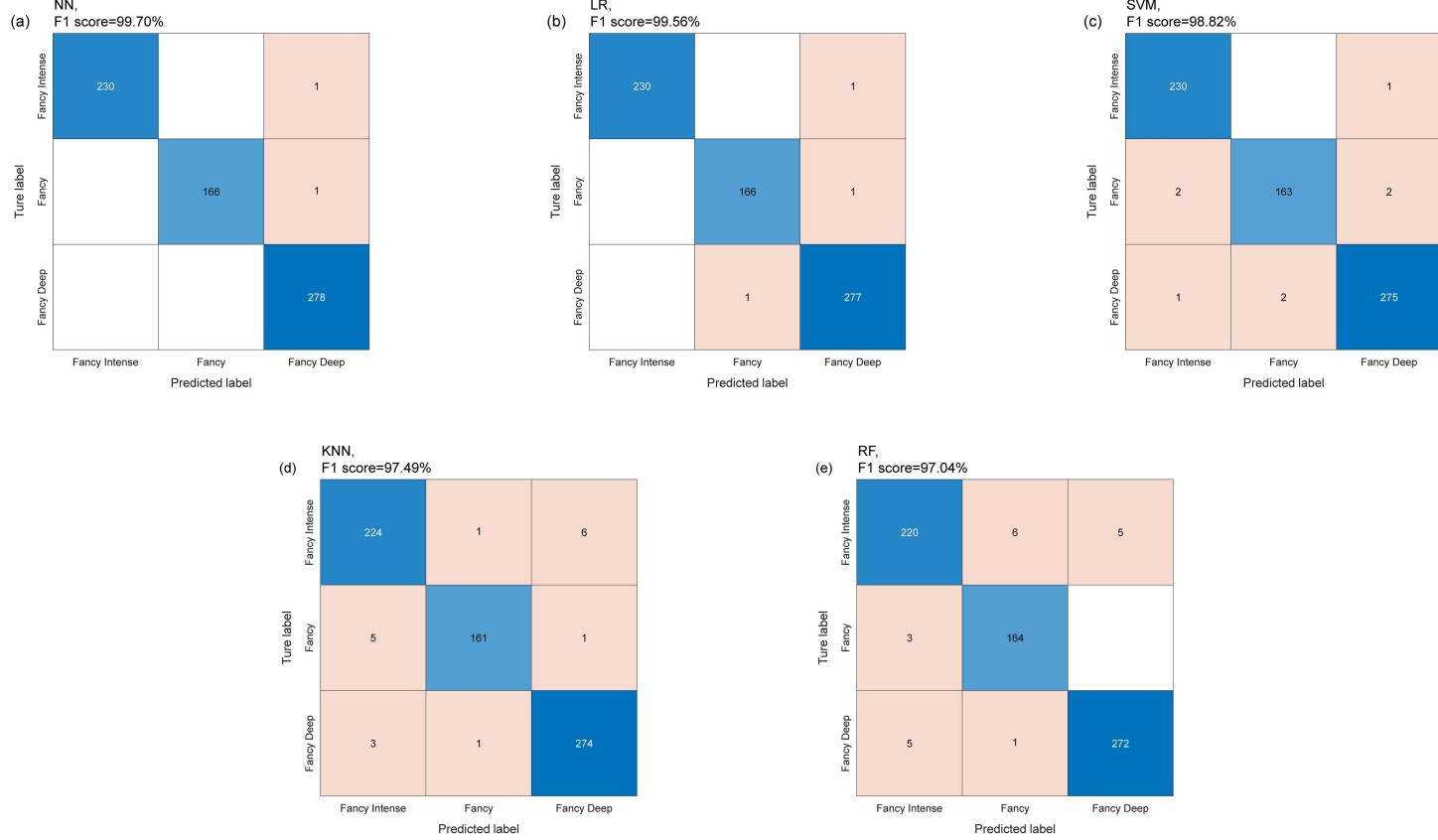

**Fig 8. Confusion matrices of the validation set for the five algorithms. (a)** LR, **(b)** NN, **(c)** SVM, **(d)** KNN, and **(e)** RF. The algorithm name and its corresponding F1 score are displayed above each panel. The numbers within the squares represent the predicted category counts.

classification accuracy reflects the intrinsic separability of the three chrysoprase color clusters in CIELAB space rather than artifacts of data partitioning. Given that the classification problem involves a low-dimensional (three-feature) colorimetric representation and well-separated clusters, logistic regression with $L^2$ regularization appears sufficient to capture the underlying decision structure without overfitting.

To enhance accessibility, a chrysoprase color grading app is provided at: https://github.com/harden2009190006/Chrysoprasecolorclassifier. In this color grading tool, users simply input the $L^*$, $a^*$, and $b^*$ values of chrysoprase, and the program outputs the corresponding simulated color along with the assigned chrysoprase color category.

## Conclusion

This study presents a machine learning–based framework for objective and consistent color grading of chrysoprase, addressing the limitations of traditional subjective evaluation. By integrating colorimetric data with supervised and unsupervised algorithms, the proposed method achieves high classification performance and offers a reproducible solution with potential applicability to other colored gemstones.

Several limitations should be acknowledged. The number of real chrysoprase samples remains relatively limited (n = 51), which may restrict the robustness of conclusions regarding broader generalizability. In addition, all samples in this study were sourced from Australia, and future studies incorporating chrysoprase from additional geographic deposits would help further evaluate the generalizability of the proposed framework.

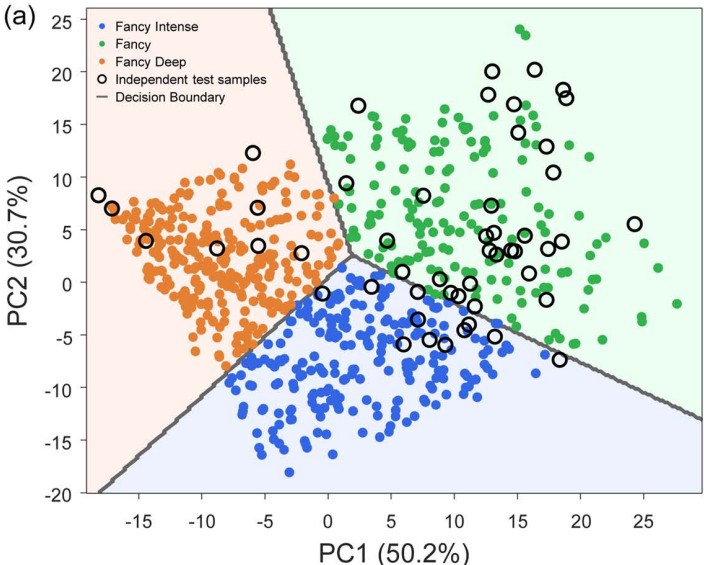
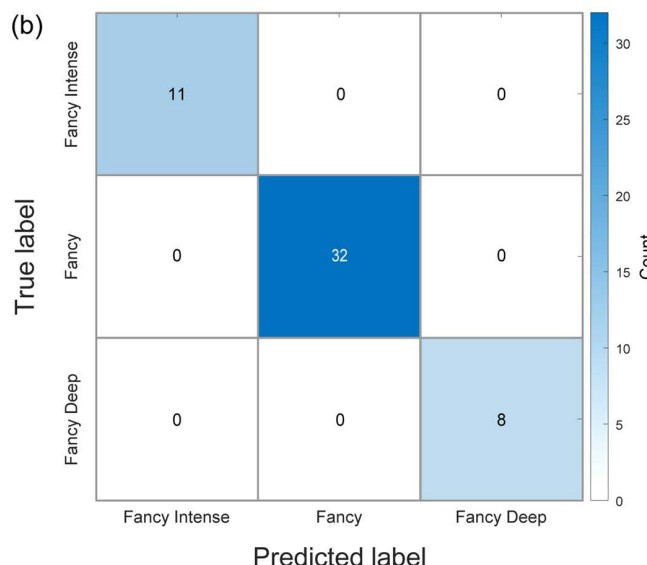

**Fig 9. Classification performance of the logistic regression model on the independent chrysoprase dataset. (a)** Logistic regression decision regions in PCA space. Background colors indicate predicted grades. Solid points represent training samples, and open circles denote independent test samples. Gray lines show decision boundaries. PC1 and PC2 correspond to the first two principal components derived from the $L^*a^*b^*$ color features and explain 50.2% and 30.7% of the total variance, respectively. **(b)** Confusion matrix for the 51 independent chrysoprase samples.

**Table 4. Per-class classification metrics for the independent chrysoprase test set.**

| Class | Precision | Recall | F1-score | Support |
|---|---|---|---|---|
| Fancy Intense | 1.00 | 1.00 | 1.00 | 11 |
| Fancy | 1.00 | 1.00 | 1.00 | 32 |
| Fancy Deep | 1.00 | 1.00 | 1.00 | 8 |
| Macro average | 1.00 | 1.00 | 1.00 | 51 |

Another limitation is that the current framework is primarily suited to gemstones with relatively uniform color distributions. For materials exhibiting complex optical phenomena—such as color zoning or pleochroism—spectrophotometric measurements alone may not fully capture perceptual appearance. Future research will therefore focus on expanding real-sample datasets and incorporating imaging-based or perception-driven approaches to enhance model robustness and extend applicability..

## Supporting information

**S1 Table. Color data of 676 GemDialogue references and 51 chrysoprase samples.**
(XLSX)

**S2 Table. Hyperparameter search space used in Bayesian optimization for the five machine learning algorithms.**
(DOCX)

**S1 Checklist. Inclusivity in global research questionnaire.**
(DOCX)

## Acknowledgments

The authors would like to thank Yu Wang, Jiayang Han, Dan Wang, Xiang Zong, and Qingfeng Guo for their valuable assistance with the experiments and insightful suggestions during the preparation of this manuscript. The experiments were conducted in the laboratory of the School of Gemmology at China University of Geosciences, Beijing.

## Author contributions

**Conceptualization:** Yuansheng Jiang, Ying Guo, Vien Cheung, Pohsun Wang, Stephen Westland.

**Data curation:** Yuansheng Jiang, Ying Guo, Vien Cheung.

**Formal analysis:** Yuansheng Jiang, Ying Guo, Vien Cheung.

**Funding acquisition:** Yuansheng Jiang, Ying Guo, Pohsun Wang.

**Investigation:** Yuansheng Jiang, Vien Cheung.

**Methodology:** Yuansheng Jiang, Ying Guo, Vien Cheung.

**Project administration:** Yuansheng Jiang, Ying Guo, Vien Cheung, Pohsun Wang.

**Resources:** Yuansheng Jiang.

**Software:** Yuansheng Jiang, Vien Cheung, Stephen Westland.

**Supervision:** Vien Cheung, Pohsun Wang, Stephen Westland.

**Validation:** Yuansheng Jiang, Vien Cheung, Pohsun Wang, Stephen Westland.

**Visualization:** Yuansheng Jiang, Ying Guo, Vien Cheung, Stephen Westland.

**Writing – original draft:** Yuansheng Jiang.

**Writing – review & editing:** Yuansheng Jiang, Ying Guo, Vien Cheung, Pohsun Wang, Stephen Westland.

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
