## [Decision Letter · Decision Letter 0]

5 Feb 2026

PONE-D-25-23974Chrysoprase color grading with machine learning: a systematic approachPLOS One

Dear Dr. guo,

Thank you for submitting your manuscript to PLOS ONE. After careful consideration, we feel that it has merit but does not fully meet PLOS ONE’s publication criteria as it currently stands. Therefore, we invite you to submit a revised version of the manuscript that addresses the points raised during the review process.

Please revise the manuscript in response to the attached reviews. Details can be found below.

We look forward to receiving your revised manuscript.

Kind regards,

Agnieszka Konys, Ph.D.

Academic Editor

PLOS One

Journal Requirements:

4. We are unable to open your Supporting Information file [S2_File.mlapp]. Please kindly revise as necessary and re-upload.

Reviewers' comments:

Reviewer's Responses to Questions

**Comments to the Author**

1. Is the manuscript technically sound, and do the data support the conclusions?

Reviewer #1: No

Reviewer #2: Yes

Reviewer #3: Partly

2. Has the statistical analysis been performed appropriately and rigorously? 

Reviewer #1: No

Reviewer #2: Yes

Reviewer #3: No

3. Have the authors made all data underlying the findings in their manuscript fully available?

Reviewer #1: No

Reviewer #2: Yes

Reviewer #3: Yes

4. Is the manuscript presented in an intelligible fashion and written in standard English?

Reviewer #1: No

Reviewer #2: Yes

Reviewer #3: Yes

5. Review Comments to the Author

Reviewer #1: I have encountered some negative comments regarding your article. I do not think it is possible to publish your article in its current form. Please review the comments carefully.

I have encountered some negative comments regarding your article. I do not think it is possible to publish your article in its current form. Please review the comments carefully.

Reviewer #2: 1. Broaden Sample Representation for Enhanced Generalizability

The study relies on 51 chrysoprase samples exclusively sourced from Australia, which limits the model’s ability to account for color variations across global deposits. Chrysoprase from regions like Tanzania, Poland, or Kazakhstan often exhibits distinct color traits due to differing geological conditions and nickel content—factors that directly influence its market valuation. Expanding the sample set to include these international sources would ensure the grading system reflects real-world diversity. Additionally, focusing solely on cabochon-cut specimens with uniform color overlooks common natural variations like subtle color zoning or minor inclusions. Incorporating such samples would test the model’s robustness in practical, non-ideal scenarios, making it more reliable for industry use.

2. Strengthen Alignment Between Model Output and Industry Practice

While the logistic regression model delivers impressive technical performance, the manuscript does not clearly connect its color grading outcomes to standard gemological workflows. Gemologists and appraisers rely on intuitive, perceptually meaningful criteria—not just numerical Lab* values—to assess color quality. For example, explaining how the “Fancy Intense,” “Fancy,” and “Fancy Deep” categories map to existing trade terminology or price differentials would bridge this gap. Supplementing with a direct comparison between the model’s results and evaluations from certified gemologists would also validate its relevance. Without this real-world context, the tool risks being seen as academically sound but impractical for day-to-day gem grading.

3. Address Practical Accessibility of the Grading Tool

The proposed color grading app requires input of Lab* values measured with an X-Rite SP62 spectrophotometer—a specialized piece of equipment most jewelers or small-scale traders do not own. This limits the tool’s widespread adoption. To enhance usability, the authors should test the model’s performance with data from more accessible devices, such as portable colorimeters or calibrated smartphone cameras. Providing guidance on how to obtain accurate color data without high-end equipment would make the tool actionable for a broader audience. Additionally, including a user-friendly interface demonstration or step-by-step guide for non-technical users would lower barriers to adoption, ensuring the research translates into tangible industry value.

4. Relevant References to Strengthen Methodological Context

To better contextualize the study’s core contribution—applying a systematic, data-driven classification framework to solve a longstanding industry challenge of subjectivity—it is recommended to cite the following two articles in the discussion section. These references align with the study’s emphasis on rigorous, cross-disciplinary methodology and standardization:

A comprehensive review on targeting diverse immune cells for anticancer therapy: Beyond immune checkpoint inhibitors（Crit Rev Oncol Hematol. 2025）This review advocates for moving beyond single-target approaches to develop multi-faceted, standardized strategies for complex biological classification—an idea directly paralleled in this study. The authors of the anticancer review highlight that relying on isolated markers (analogous to traditional subjective gem color assessment) leads to inconsistent outcomes, while integrating multiple data streams and systematic validation (like the combination of K-means clustering, Fisher discriminant analysis, and Bayesian optimization here) enhances reliability. Citing this work reinforces the broader significance of the current study’s methodology: just as the anticancer field benefits from standardized, multi-dimensional classification, gemology gains from objective, data-integrated grading that reduces human bias. This cross-disciplinary connection underscores the generalizability of systematic classification frameworks across fields with subjective evaluation challenges.

The role of cisplatin in modulating the tumor immune microenvironment and its combination therapy strategies: a new approach to enhance anti-tumor efficacy（Ann Med. 2025）This article focuses on optimizing therapeutic outcomes through strategic method combination and validation—an objective shared by the current chrysoprase grading study. The anticancer research emphasizes that combining complementary tools (e.g., drug therapy with immune modulation) addresses limitations of single-method approaches, much like how this study integrates colorimetric data, unsupervised clustering, and supervised machine learning to overcome the flaws of manual color grading. Additionally, the article stresses the importance of translating technical advancements into practical, standardized tools—mirroring this study’s development of a publicly accessible grading app. Citing this reference strengthens the discussion of the current work’s impact: both studies demonstrate how rigorous method integration and practical tool development can transform fields reliant on inconsistent, subjective assessments, making the gemology framework more relatable to broader scientific discourse on standardization.

Reviewer #3: 1.The training set is composed entirely of 676 synthetic GemDialogue-derived points while the test set is only the 51 real chrysoprase samples. This design risks domain mismatch (synthetic vs real) and over-optimistic generalization claims. The perfect classification of the 51 samples by the final LR model is surprising and may reflect leakage, overly narrow class boundaries, or lack of realistic variance in the training set. Please (a) clarify the rationale for treating all real samples as a pure test set, and (b) repeat model evaluation using cross-validation schemes that incorporate the real samples into training folds (for example: stratified k-fold where both synthetic and real samples are mixed during fold assignment, or leave-one-out on real samples). Report results of these additional evaluations and discuss differences.

2.procedure to create 2,730 overlay colors and then filter to 676 representative points requires more quantitative justification. Describe precisely (and provide code snippets or parameters) how the filtering used the L*, C*, h° bounds of the 51 samples. Discuss whether GemDialogue overlays realistically simulate cabochon optics (translucency, surface reflection, light scattering). If possible, validate the synthetic set by comparing distributions (e.g., kernel density or convex hull) of L*a*b* values between synthetic and measured samples. If synthetic data do not adequately cover real-world variability, consider augmenting the training set with measured variations (different orientations, multiple measurement locations per cabochon, different polish conditions, inclusion of proximate non-ideal samples).

3.n = 51 real samples is small for definitive claims. Provide confidence intervals or statistical tests on classifier performance (e.g., bootstrap on the 51 samples, or uncertainty estimates from repeated resampling). Report per-class sample counts for the real samples and indicate whether any class is under-represented; if so, show class-wise precision/recall on the test set. The manuscript currently reports only overall macro F1 and an assertion that all 51 were correctly assigned; show the confusion matrix and per-class metrics for the test set and provide explanation if classes have very small n.

4.The number of clusters (three) was chosen using silhouette applied to only the 51 real samples; then k-means with k=3 was applied to the combined dataset to label synthetic points. This approach is reasonable but may bias labeling toward the real-sample distribution. Make explicit the assumptions and potential limitations of this choice. Also, adopting GIA “Fancy” nomenclature (developed for diamonds) for chrysoprase requires caution — justify this transfer more explicitly (e.g., perceptual studies or prior literature applying similar labels to coloured gemstones).

5.Provide exact measurement geometry (you state SCI and D65, 2° observer) but clarify: measurement aperture size, number and location of measurement points on each cabochon (top only or multiple locations?), orientation control, whether measurements were taken through a mounting (e.g., on white card) or handheld, and repeatability (report standard deviation of triplicate measurements). Also indicate whether measurements were made on faceted or cabochon surfaces (you state cabochons but shapes differ); polishing and curvature affect measured color — discuss.

6.Hyperparameters table is useful, but please provide (in methods or supplementary) the optimization ranges used for Bayesian optimization, the Bayesian optimizer settings (acquisition function, iterations), and the random seed(s). Confirm whether reported results are deterministic across seeds. Provide code or a README in the GitHub repo explaining how to reproduce training and evaluation exactly (including MATLAB versions and toolboxes used).

7.The manuscript shows validation confusion matrices (via CV) but does not present the confusion matrix for the independent 51-sample test set other than stating perfect assignment. Include: (a) full confusion matrix for the 51 samples, (b) per-class precision/recall/F1, (c) visualization of decision boundaries in L*a*b* (or PCA) with real samples highlighted, and (d) calibration plots or probability outputs (for logistic regression) to assess classifier confidence.

6. PLOS authors have the option to publish the peer review history of their article (what does this mean?). If published, this will include your full peer review and any attached files.

Reviewer #1: No

Reviewer #2: No

Reviewer #3: **Yes:** Dr. S M Rashidul Hasan

---

## [Author Response · Author response to Decision Letter 1]

20 Apr 2026

Dear Dr. Konys,

Thank you very much for the opportunity to revise our manuscript. We appreciate the constructive comments and suggestions provided by you and the reviewers.

We will carefully address all the points raised during the review process and revise the manuscript accordingly. The revised manuscript and the detailed response to the reviewers will be submitted through the system within the requested timeframe.

Thank you again for your consideration of our work.

Kind regards,

Yuansheng Jiang

Reviewer #1

I have encountered some negative comments regarding your article. I do not think it is possible to publish your article in its current form. Please review the comments carefully.

Response: We respectfully clarify that the scientific content of this manuscript, including the experimental design, data collection, data analysis, and interpretation of the results, was entirely developed and written by the authors based on their original research. The study is grounded in experimental measurements of chrysoprase samples and independent data analysis conducted by the research team.

The manuscript was initially drafted by the authors in their native language. To improve the clarity and readability of the English text, AI-assisted tools were used only for translation and language polishing. These tools were applied solely to refine grammar and wording. They were not used to generate scientific ideas, design experiments, analyze data, or produce the scientific conclusions presented in the manuscript.

The authors fully understand the importance of academic integrity in scientific publishing and take complete responsibility for the originality, accuracy, and authenticity of the work reported in this manuscript.

Reviewer #2:

1. Broaden Sample Representation for Enhanced Generalizability

The study relies on 51 chrysoprase samples exclusively sourced from Australia, which limits the model’s ability to account for color variations across global deposits. Chrysoprase from regions like Tanzania, Poland, or Kazakhstan often exhibits distinct color traits due to differing geological conditions and nickel content—factors that directly influence its market valuation. Expanding the sample set to include these international sources would ensure the grading system reflects real-world diversity. Additionally, focusing solely on cabochon-cut specimens with uniform color overlooks common natural variations like subtle color zoning or minor inclusions. Incorporating such samples would test the model’s robustness in practical, non-ideal scenarios, making it more reliable for industry use.

Response: We sincerely thank the reviewer for this valuable suggestion regarding the representativeness of the chrysoprase dataset and the applicability of the proposed grading model to natural gemstones.

(1) Geographic origin of samples

We agree that chrysoprase from different deposits (e.g., Tanzania, Poland, and Kazakhstan) may show variations in color characteristics due to differences in geological environments and nickel distribution. Expanding the dataset to include a broader range of geographic sources would indeed be beneficial for evaluating the global generalizability of the proposed grading framework.

However, it is important to note that reliable geographic origin determination for chrysoprase remains challenging. Previous mineralogical and isotopic studies have attempted to distinguish chrysoprase from different deposits, but these investigations generally report no clear isotopic signatures that allow consistent discrimination between geographic sources [1]. As a result, chrysoprase origin determination is currently not routinely achievable in gemological practice.

In this study, Australian chrysoprase was selected primarily because it represents one of the most widely studied and commercially important sources and exhibits the characteristic apple-green color range associated with chrysoprase. The primary goal of this work is therefore to establish a methodological framework for machine-learning-based chrysoprase color grading, rather than to construct a geographically comprehensive grading standard. We agree that future work should incorporate samples from additional deposits to further evaluate the robustness of the model.

(2) Use of uniformly colored cabochon samples

We also appreciate the reviewer’s comment regarding the use of uniformly colored cabochon specimens. In the present study, samples with relatively uniform color were intentionally selected due to limitations of the spectrophotometric measurement method. Color measurements were conducted using an X-Rite SP62 spectrophotometer with a 5 mm measurement aperture, which records reflectance spectra from a single measurement area. When strong color zoning or heterogeneous inclusions occur within the measurement spot, the recorded spectrum represents a spatial average of multiple color regions and may not accurately represent any specific perceived color of the gemstone.

To ensure that the measured CIELAB values correspond reliably to the perceived color of the specimens, relatively homogeneous cabochon samples were therefore used in this initial study. Establishing the grading model under controlled color conditions allows the relationship between colorimetric parameters and classification results to be evaluated more clearly.

We agree that natural chrysoprase may exhibit color zoning or inclusions in practical gemological contexts. Incorporating such samples would be valuable for testing the robustness of the grading system under real-world conditions. Future research will therefore aim to include more heterogeneous specimens and explore multi-point spectrophotometric measurements or imaging-based color analysis, which can better capture spatial color variation. This limitation and the corresponding future research directions have also been clarified in the Conclusion section of the revised manuscript.

2. Strengthen Alignment Between Model Output and Industry Practice

While the logistic regression model delivers impressive technical performance, the manuscript does not clearly connect its color grading outcomes to standard gemological workflows. Gemologists and appraisers rely on intuitive, perceptually meaningful criteria—not just numerical Lab* values—to assess color quality. For example, explaining how the “Fancy Intense,” “Fancy,” and “Fancy Deep” categories map to existing trade terminology or price differentials would bridge this gap. Supplementing with a direct comparison between the model’s results and evaluations from certified gemologists would also validate its relevance. Without this real-world context, the tool risks being seen as academically sound but impractical for day-to-day gem grading.

Response: We thank the reviewer for this insightful comment regarding the connection between the proposed grading framework and practical gemological workflows.

At present, chrysoprase does not have a widely accepted standardized color grading system in gemological practice, and color evaluation is largely based on subjective visual judgment. One motivation of this study is therefore to explore whether colorimetric measurements combined with machine learning can provide a more objective framework for describing chrysoprase color variation. The CIELAB color space used in this study was specifically designed to approximate perceptual color differences and is widely used in color science to quantitatively represent human color perception.

To facilitate communication with gemological practice, the three clusters identified in this study were assigned descriptive names—“Fancy Intense,” “Fancy,” and “Fancy Deep”—referencing the widely used GIA terminology for colored diamonds. These terms are intended as descriptive labels for relative lightness–chroma differences rather than as a direct application of the diamond grading system.

Regarding the reviewer’s suggestion to relate the grading results to market price, chrysoprase currently lacks a transparent or standardized pricing system, as prices are typically determined through private trade and influenced by multiple factors beyond color.

The reviewer also suggested comparing the model results with evaluations from certified gemologists. While such comparisons would certainly be valuable, conducting a structured expert evaluation requires carefully controlled viewing conditions and dedicated perceptual experiments, which are beyond the scope of the present study. Nevertheless, the proposed framework provides a reproducible and objective method for describing chrysoprase color variation, which may serve as a useful reference for future development of practical grading systems. This aspect will be considered in future work.

3. Address Practical Accessibility of the Grading Tool

The proposed color grading app requires input of Lab* values measured with an X-Rite SP62 spectrophotometer—a specialized piece of equipment most jewelers or small-scale traders do not own. This limits the tool’s widespread adoption. To enhance usability, the authors should test the model’s performance with data from more accessible devices, such as portable colorimeters or calibrated smartphone cameras. Providing guidance on how to obtain accurate color data without high-end equipment would make the tool actionable for a broader audience. Additionally, including a user-friendly interface demonstration or step-by-step guide for non-technical users would lower barriers to adoption, ensuring the research translates into tangible industry value.

Response: We thank the reviewer for this valuable comment regarding the practical accessibility of the proposed color grading tool.

It should be clarified that the X-Rite SP62 spectrophotometer used in this study is a portable handheld instrument designed for field measurements, and is widely used in industrial color quality control and colorimetric research.

In gemological practice, objective color grading and certification are typically performed by professional gemological laboratories or third-party certification institutions, where calibrated instruments and standardized procedures are used to ensure reliable and reproducible results. In this context, the use of a spectrophotometer for colorimetric measurement is consistent with current professional practices for gemstone evaluation.

We nevertheless agree that exploring alternative approaches for obtaining colorimetric data from consumer imaging devices, such as calibrated smartphone cameras, could further expand the accessibility of the proposed framework. However, achieving reliable colorimetric accuracy with such devices remains challenging due to variations in illumination conditions, camera sensors, and calibration procedures. This represents an interesting direction for future research.

Regarding usability, the proposed grading tool has been implemented as a publicly accessible application, where users only need to input the measured L*, a*, and b* values to obtain the corresponding color category, keeping the workflow straightforward once colorimetric data are available.

4. Relevant References to Strengthen Methodological Context

To better contextualize the study’s core contribution—applying a systematic, data-driven classification framework to solve a longstanding industry challenge of subjectivity—it is recommended to cite the following two articles in the discussion section. These references align with the study’s emphasis on rigorous, cross-disciplinary methodology and standardization:

A comprehensive review on targeting diverse immune cells for anticancer therapy: Beyond immune checkpoint inhibitors（Crit Rev Oncol Hematol. 2025）This review advocates for moving beyond single-target approaches to develop multi-faceted, standardized strategies for complex biological classification—an idea directly paralleled in this study. The authors of the anticancer review highlight that relying on isolated markers (analogous to traditional subjective gem color assessment) leads to inconsistent outcomes, while integrating multiple data streams and systematic validation (like the combination of K-means clustering, Fisher discriminant analysis, and Bayesian optimization here) enhances reliability. Citing this work reinforces the broader significance of the current study’s methodology: just as the anticancer field benefits from standardized, multi-dimensional classification, gemology gains from objective, data-integrated grading that reduces human bias. This cross-disciplinary connection underscores the generalizability of systematic classification frameworks across fields with subjective evaluation challenges.

The role of cisplatin in modulating the tumor immune microenvironment and its combination therapy strategies: a new approach to enhance anti-tumor efficacy（Ann Med. 2025）This article focuses on optimizing therapeutic outcomes through strategic method combination and validation—an objective shared by the current chrysoprase grading study. The anticancer research emphasizes that combining complementary tools (e.g., drug therapy with immune modulation) addresses limitations of single-method approaches, much like how this study integrates colorimetric data, unsupervised clustering, and supervised machine learning to overcome the flaws of manual color grading. Additionally, the article stresses the importance of translating technical advancements into practical, standardized tools—mirroring this study’s development of a publicly accessible grading app. Citing this reference strengthens the discussion of the current work’s impact: both studies demonstrate how rigorous method integration and practical tool development can transform fields reliant on inconsistent, subjective assessments, making the gemology framework more relatable to broader scientific discourse on standardization.

Response: We thank the reviewer for the suggestion to include additional references to strengthen the methodological context of this study.

The recommended articles focus on anticancer therapy and tumor immunology, which belong to a substantially different research field from the present work in gemology and colorimetric classification. While we appreciate the reviewer’s intention to highlight potential cross-disciplinary methodological parallels, the specific scientific contexts and research objectives of these studies differ considerably from the scope of the current manuscript.

Therefore, these references were not directly incorporated into the manuscript. Instead, we have further clarified the methodological framework of the present study and its contribution to objective and data-driven color classification in gemology in the discussion section.

Reviewer #3:

1. The training set is composed entirely of 676 synthetic GemDialogue-derived points while the test set is only the 51 real chrysoprase samples. This design risks domain mismatch (synthetic vs real) and over-optimistic generalization claims. The perfect classification of the 51 samples by the final LR model is surprising and may reflect leakage, overly narrow class boundaries, or lack of realistic variance in the training set. Please (a) clarify the rationale for treating all real samples as a pure test set, and (b) repeat model evaluation using cross-validation schemes that incorporate the real samples into training folds (for example: stratified k-fold where both synthetic and real samples are mixed during fold assignment, or leave-one-out on real samples). Report results of these additional evaluations and discuss differences.

Response: We sincerely thank the reviewer for raising this important concern regarding the potential domain mismatch between synthetic and real samples.

(a) Rationale for using real samples as a pure test set

The primary objective of this study is to develop a color grading model that can generalize to unseen chrysoprase samples. The 676 GemDialogue-derived color points were designed to systematically simulate the perceptual green color space relevant to chrysoprase, thereby forming a comprehensive calibration domain. The 51 natural chrysoprase samples were therefore reserved as an external test set to evaluate the model’s performance on real-world data.

This design intentionally follows a calibration–vali

---

## [Editor Report · Decision Letter 1]

27 Apr 2026

Chrysoprase color grading with machine learning: a systematic approach

PONE-D-25-23974R1

Dear Dr. Cheung,

We’re pleased to inform you that your manuscript has been judged scientifically suitable for publication and will be formally accepted for publication once it meets all outstanding technical requirements.

Kind regards,

Agnieszka Konys, Ph.D.

Academic Editor

PLOS One
---

## [Editor Report · Acceptance letter]

PONE-D-25-23974R1

PLOS One

Dear Dr. Cheung,

I'm pleased to inform you that your manuscript has been deemed suitable for publication in PLOS One. Congratulations! Your manuscript is now being handed over to our production team.

Kind regards,

on behalf of

Dr. Agnieszka Konys

Academic Editor

PLOS One